# A New Species of the Mealybug Genus *Mirococcus* (Hemiptera: Coccomorpha: Pseudococcidae) from the Cape Verde Islands, with New Records and an Updated Checklist of Scale Insect Species

**DOI:** 10.3390/insects13110999

**Published:** 2022-10-31

**Authors:** Bożena Łagowska, Katarzyna Golan, Christopher J. Hodgson

**Affiliations:** 1Department of Plant Protection, University of Life Sciences in Lublin, Leszczyńskiego 7, 20-069 Lublin, Poland; 2Department of Biodiversity and Biological Systematics, The National Museum of Wales, Cardiff CF10 3NP, UK

**Keywords:** coccoids, fauna, list, locality data, distribution

## Abstract

**Simple Summary:**

In this study, a new mealybug species from the Cape Verde Islands, *Mirococcus capeverdensis* Łagowska and Hodgson sp. n., collected on *Campylanthus glaber* Benth. (Scrophulariaceae) is described and illustrated based on the adult female. A key to the mealybugs from the Afrotropical Region that lack cerarii is provided. In addition, seven scale insect species are recorded for the first time from the Cape Verde Islands. An updated checklist of Coccomorpha species from this region, along with their known island distributions and valid sources, is also appended. Prior to this study, 38 scale insect species in 7 families and 28 genera are known from the Cape Verde Islands.

**Abstract:**

In this study, a new species of mealybug from the Cape Verde Islands, *Mirococcus capeverdensis* Łagowska and Hodgson sp. n., collected on *Campylanthus glaber* Benth. (Scrophulariaceae)***,*** a native plant to these islands, is described and illustrated based on the adult female. A taxonomic key to the mealybugs from the Afrotropical Region that lack cerarii is provided. In addition, seven scale insect species, namely *Aulacaspis tubercularis* Newstead, *Hemiberlesia cyanophylli* (Signoret), *Pseudaonidia trilobitiformis* (Green), *Icerya aegyptiaca* (Douglas), *Maconellicoccus hirsutus* (Green), *Palmicultor palmarum* (Ehrhorn), and *Pseudococcus comstocki* (Kuwana) are recorded for the first time from the Cape Verde Islands and an updated checklist of Coccomorpha species known from this region, along with their known island distributions and valid sources, is appended.

## 1. Introduction

The Afrotropical region has a unique and diverse scale insect fauna with 1458 species distributed on the African continent and its many oceanic islands, including the Cape Verde archipelago [1].

The Cape Verde volcanic archipelago is a group of 10 major islands plus several smaller uninhabited islets located in the central Atlantic Ocean, approximately 570 km off the western coast of the African continent, to the west of Senegal, Gambia, and Mauritania. It lies between 14° to 18° N latitude and 22° to 26° W longitude and forms part of the Macaronesia ecoregion, along with the Azores, the Canary Islands, Madeira, and the Savage Isles. The total land area is approximately 4033 km^2^, with the islands of Santiago (785.0 km^2^) and Santo Antão (785 km^2^) being the largest. Three of the Cape Verde islands, Sal, Boa Vista, and Maio, are fairly flat, sandy, and dry; the others are generally rockier with more vegetation. Cape Verde’s climate is milder than that of the African mainland, with average daily high temperatures ranging from 26 °C in February to 31 °C in September, whilst the average annual rainfall ranges from less than 100 mm in the arid coastal areas to more than 1000 mm in the humid mountains. Vegetation in the islands is similar to that of the savannah or steppe type. Some of the plant species are considered to be endemic, having evolved over millions of years of isolation [2,3].

According to ScaleNet, an open-source database for scale insects [1], prior to this study, 38 scale insect species in 7 families and 28 genera had been recorded from the Cape Verde Islands. These data were the result of studies by only a few scientists, notably Fernandes [4,5,6,7], Schmutterer et al. [8], and Van Harten et al. [9]. In February 2018, the first author had the opportunity to travel to three of the Cape Verde islands (Sal, Santo Antão, and São Vicente) and collected scale insects of 5 families and 18 genera, among them an undescribed species, *Mirococcus capeverdensis* Łagowska and Hodgson sp. n.

In this paper, we describe and illustrate the adult female of this new species of *Mirococcus*, which was collected on *Campylanthus glaber* Benth. (Scrophulariaceae), a native plant to the Cape Verde Islands. In addition, we include a key to the mealybugs known from the Afrotropics that lack cerarii. Of the other species collected, seven are new species records for the Cape Verde Islands and an updated checklist of Coccomorpha from this archipelago is also included.

## 2. Materials and Methods

The material studied was collected during February 2018 on Sal, Santo Antão, and the São Vicente islands, mainly on wild plants. In total, approximately 100 lots of scale insect material were collected and studied to select suitable specimens for slide-mounting. These were stained and mounted in Canada Balsam on glass slides, mainly using the methodology in Hodgson and Henderson [10], except the specimens were left in cold KOH for one or two days before clearing. The figure of *M. capeverdensis* Łagowska and Hodgson sp. n. shows the dorsum on the left side and the venter on the right, with vignettes of important structures (not drawn to scale) around the margin. The measurements provided for the legs, spiracles, clypeolabral shield, and anal plates are for their greatest width or length. The morphological terms for Pseudococcidae follow those of Williams [11].

The slides are stored in the Department of Plant Protection, at the University of Life Sciences in Lublin, except for the holotype and paratype of the new species, which were deposited in the Natural History Museum, Cromwell Road, London, UK (BMNH).

## 3. Results


**Taxonomy**

**Pseudococcidae**


*Mirococcus* Borchsenius, 1947: 142 [12]**Type species**: *Phenacoccus inermis* Hall, 1925 by original designation.

**Generic diagnosis.** Members of the genus *Mirococcus* can be diagnosed by the following combination of features: antennae eight or nine segmented (rarely seven); legs usually normally developed; claw usually with a denticle; anal apparatus complete in type species, but much simplified in other species; posterior pair of ostioles always present, anterior pair sometimes absent; multilocular and trilocular pores present; quinquelocular pores present or absent; oral collar tubular ducts usually simple; oral rim ducts absent; cerarii absent, and conical setae absent [13].**Comment.***Mirococcus* is one of the few genera of mealybugs that completely lack cerarii and conical setae. Species found within the African continent lacking cerarii can be separated using the key below.

***Mirococcus capeverdensis*** Łagowska and Hodgson sp. n.

**Material studied**: **Holotype:** Left label: **Cape Verde Is /** Sal, St. Maria / *Campylanthus glaber* / 14.ii.2018 / B. Łagowska; right label: *Mirococcus / capeverdensis /* Łagowska & / Hodgson / Holotype (one adult female in fair–good condition, body slightly twisted). **Paratype:** Data are the same as for the holotype, a single adult female but broken in two, with each half mounted on separate slides. The anterior half is poor, and the posterior half is good. All three slides were deposited in BMNH, London.

**Description of adult female** (Figure 1). Data for paratype in brackets. Appearance in life not recorded. Mounted adult female 1.3 mm long and 0.5 mm wide. Anal lobes small, each with a long apical seta 118–126 (118) µm long, plus several other setae, each 50–76 (55) µm long; anal lobe bar absent. Antennae eight (nine) segmented, each approximately 223 (252) µm long. Legs well developed, with lengths (iii) in µm: coxa 90–95 (116–120), trochanter + femur 168 (189); tibia 144 (151), and tarsus 84–86 (84); tarsal digitules setose; claw approximately 26–28 (27) um long, with a strong denticle; claw digitules knobbed, longer than claw; translucent pores present on distal end of femur and on tibia. Labium approximately 100 µm long; clypeolabral shield with a single pair of long setae. Spiracles with anterior peritreme 25 (32–34) µm wide, posterior peritreme 32 (42) µm wide. Circulus well developed, approximately 50 (92) µm wide, possibly divided by an intersegmental line. Ostioles present, with few trilocular pores on lips and no setae. Cerarii absent but with very long setose setae (lengths ranging from 35 µm to 60 µm) in groups of one to three along margin, located approximately segmentally. Each group also with one to three shorter setae, each approximately 25–35 µm long.

Dorsal surface with rows of setae, some very short (only about 8 µm long) but also with a few setae similar to long marginal setae, each 40–50 (40–50) µm long with a few on all segments. Multilocular disc pores, each 6–8 µm wide, frequent throughout but most abundant posteriorly; where present mainly along posterior margin of each segment; those posteriorly all with 10 loculi, but number of loculi decreasing more anteriorly, with five or six loculi in some smaller pores. Quinquelocular pores otherwise absent. Trilocular pores slightly larger than usual, almost round, each 3–4 µm wide, and present throughout. Simple pores, each approximately 1.5 µm wide, present sparsely throughout. Oral collar tubular ducts, each 8–10 µm long, present throughout but most abundant posteriorly and marginally. Oral rim ducts absent.

Ventral surface with mainly slender setose setae but with an occasional quite long seta, as on dorsum. Multilocular disc pores as on dorsum, throughout but most abundant on abdomen; where present mainly along both anterior and posterior margins. Trilocular pores, as on dorsum, evenly distributed. Simple pores as on dorsum. Oral collar tubular ducts present, as on dorsum, but more abundant and present more or less throughout. Oral rim ducts absent.

**Etymology.** The name of the species is derived from Cape Verde Islands, the archipelago from which this species was collected, with the adjectival suffix -*ensis*, indicating the place of origin.

**Comment.** Adult female *M. capeverdensis* sp. n. can be diagnosed by a combination of the following features: (i) all cerarii and conical setae absent; (ii) dorsum, margin, and venter with very long setae, each up to 60 µm long; (iii) small anal lobes; (iv) legs with translucent pores; (v) each claw with a strong denticle; (vi) multilocular pores present on both dorsum and venter; (vii) oral collar tubular ducts all about the same size, present on dorsum and venter, and; (viii) trilocular pores slightly larger than usual, almost round. 

*M. capeverdensis* sp. n. differs from all other species in this genus in having very long setae on the dorsum and margin.

Of the species currently included in *Mirococcus*, adult female *M. capeverdensis* sp. n. are most similar to that of *M. inermis* (Hall), sharing the following features: (i) cerarii and conical setae absent; (ii) circulus well developed; (iii) both anterior and posterior ostioles present; (iv) legs with translucent pores; (v) simple pores scattered on the dorsum and venter; (vi) multilocular pores present on the dorsum and venter, and (vii) oral collar tubular ducts of one size present on the dorsum and venter. However, it differs as follows (character-states in *M. inermis* in brackets): (i) many dorsal setae, very long and setose (all dorsal setae short and slender), (ii) claw digitules capitate (claw digitules setose), and (iii) claw with a strong denticle (claw denticle weak).

With the description of *M*. *capeverdensis* sp. n., the total number of species in *Mirococcus* is increased to 15, distributed mainly in the Palaearctic Region (14 species). Currently, only two *Mirococcus* species are known from the Afrotropics: *M. inermis* (Hall), recorded from north Africa (Egypt, Sudan, and Tunisia), and *M. capeverdensis* sp. n. from the Cape Verde Islands.

**Key to mealybug genera and species in the Afrotropical Region that lack cerarii, including cerarii on the anal lobes (mainly after Watson (unpublished)** [14] **and Millar** [15]**).**

1.Dorsal setae and ventral marginal body setae very stoutly spiniform with apices rounded or truncated, and some slightly curved. Antennae nine segmented. Hind leg without translucent pores. Claw with a denticle. Quinquelocular pores present … ***Eriocorys hystrix*** De Lotto (South Africa).

-Body setae are usually slender, with acute apices. Other characteristics not in this combination … 2

2.Multilocular disc pores absent from both dorsum and venter … 3

-Multilocular disc pores present, at least on the venter near the vulva … 5

3.Quinquelocular pores present on both the venter and dorsum. Antennae, each with six segments. Claw without a denticle … ***Lacombia*** Goux (two species: ***L. bouhelieri*** (Goux) (Morocco), ***L. dactyloni*** (Bodenheimer) (Tunisia)).

-Quinquelocular pores are absent. Antennae, each with 5–7 segments. Claw with or without a denticle … 4

4.Anal ring situated at body apex, bearing eight setae. Antenna with five or six segments. Hind leg fairly slender, without translucent pores. Found living underground in an ant’s nest … ***Bimillenia plagiolepicola*** Matile-Ferrero and Ben-Dov (Algeria).

-Anal ring situated on dorsum, bearing six setae. Antenna with seven segments. Hind leg is very robust, with translucent pores on coxa, trochanter, and femur. Found living with ants in above-ground domatia on *Acacia* … ***Acaciacoccus hockingi*** Williams and Matile-Ferrero (Kenya, Tanzania).

5.Hind coxa enlarged basally, bearing granular bosses. Margins of posterior-most four or five abdominal segments bearing lanceolate spines not grouped into cerarii. Anal opening funnel-shaped, and anal ring recessed … ***Grewiacoccus gregalis*** Brain (South Africa).

-Hind coxa not enlarged and without granular bosses. Other characteristics not as above … 6

6.Multilocular disc-pores frequent throughout dorsum and venter … 7

-Multilocular disc-pores, if present, restricted to abdominal segments only … 8

7.Claw with a strong denticle. Many dorsal setae very long and setose. Claw digitules capitate … ***Mirococcus capeverdensis*** Łagowska and Hodgson sp. n. (Cape Verde Islands).

-Claw with a weak denticle. All dorsal setae short and slender. Claw digitules not capitate … ***Mirococcus inermis*** (Hall) (Egypt, Tunisia, Sudan).

8.Anal ring without setae and with only 0–2 cells. Each trochanter with three sensilla on each surface. Claw without a denticle … ***Lenania*** De Lotto (two species: ***L. africana*** (Brain), ***L. prisca*** De Lotto (South Africa)).

-Anal ring with setae, with or without cells. Each trochanter with two sensilla on each surface. Claws with or without a denticle … 9

9.Anal lobes are well developed, each with a sclerotized area dorsally. Circulus present … ***Madeurycoccus bicolor*** De Lotto (South Africa).

-The anal lobes are undeveloped. Circulus absent … 10

10.Anterior ostioles absent. Oral rim ducts absent … 11

-Anterior ostioles present. Oral rim ducts present, at least ventrally on head … 12

11.Anal ring is complete, with a single row of cells and six anal ring setae. Multilocular disc pores absent from dorsum. Antennae each six segmented. Claw digitules setose … ***Mirococcopsis salsolae*** (Vayssière) (Tunisia).

-Anus reduced to a sclerotized ring lacking cells and setae. Multilocular disc pores present on both dorsum and venter of abdomen. Antennae each seven segmented. Claw digitules capitate … ***Mirococcopsis ptilura*** Gavrilov-Zimin (South Africa).

12.Oral rim ducts numerous on both dorsal and ventral surfaces. Antennae each six segmented. Anal ring complete, with cells and setae, the latter equal in length to diameter of anal ring. Translucent pores present on hind coxae only … ***Humococcus (Mirococcopsis) mackenziei*** Ezzat (Egypt).

-Oral rim ducts, if present on both dorsal and ventral surfaces, never numerous. Antennae each seven or eight segmented. Anal ring poorly developed, without cells. Translucent pores either absent or more widespread on hind legs … 13

13.Trilocular pores of two sizes present, with larger pores forming groups along margin. Multilocular disc pores absent from dorsum. Antennae each seven segmented. Oral rim ducts present on dorsum only. Translucent pores present on hind coxa, femur, and tibia … ***Iberococcus gomezmenori*** Matile-Ferrero (Tunisia).

-Trilocular pores of only one size present, never forming groups along margin. Multilocular disc pores present dorsally and ventrally on abdomen. Antennae each eight segmented. Oral rim ducts very few and restricted to ventrally on head. Translucent pores absent from hind legs … ***Mirococcopsis sphaerica*** (Balachowsky) (Algeria).


**Checklist of Coccomorpha from Cape Verde Islands**


In addition to the new species described above, the samples of scale insects collected on the Cape Verde Islands in 2018 by the first author also included 7 species new to the fauna and 11 species which had been reported earlier by Fernandes [4,5,6,7], Schmutterer et al. [8], and Van Harten et al. [9].

The species new to the Cape Verde islands in the list below include notes regarding their economic importance and worldwide distribution. In addition, an updated checklist of Coccomorpha species known from this archipelago, along with validation sources, is appended (Table 1). Families and species within each family are listed in alphabetical order according to the classification used in the ScaleNet database [1]. The references to species recorded from the Cape Verde Islands reported in ScaleNet have been checked and, where erroneous, corrected in the present checklist. 


**Annotated list of scale insect species newly recorded for the Cape Verde Islands**


**Family Diaspididae** Targioni Tozzetti, 1868

***Aulacaspis tubercularis*** (Newstead, 1906)Material examined: Santo Antão, 6.ii.2018, 7.ii.2018; 5 ♀♀ on the upper leaf surface of mango trees (Anacardiaceae). Distribution: A cosmopolitan species endemic to the Asian continent and introduced in other parts of the world together with infested plant material [16].***Hemiberlesia cyanophylli*** (Signoret, 1869)Material examined: Sal, Santa Maria, 14.ii.2018, ♀♀ common on different wild dicotyledonous plants, including *Heliotropium curassavicum* L. (Boraginaceae), *Synedrella nodiflora* (L.) Gaertn. (Asteraceae), and *Tetraena fontanesii* (Webb and Berthel.) (Zygophyllaceae). Distribution: A cosmopolitan, highly polyphagous species widely distributed in the tropical and subtropical regions [17,18,19,20]. ***Pseudaonidia trilobitiformis*** (Green, 1896)Material examined: São Vicente, Mindelo, 10.ii.2018, 4 ♀♀ on the upper surface of *Nerium oleander* (Apocynaceae). Distribution: A cosmopolitan species reported in 77 countries in different parts of the world [1]. Its origin is probably southern Asia, where it is common, but it has spread into Africa, the Malagasian area, and tropical America [20]. 

**Family Monophlebidae** Morrison, 1927

***Icerya aegyptiaca*** (Douglas, 1890)Material examined: Sal, Santa Maria, 11.ii.2018, 12.ii.2018, 6 ♀♀ on *Euphorbia* sp. (Euphorbiaceae); 14.ii.2018, 2♀♀ on *Tamarix senegalensis* DC. (Tamaricaceae); 15.ii.2018, 1 ♀ on *Hibiscus* sp. (Malvaceae).Distribution: A highly polyphagous species, recorded from the Afrotropical, Australasian, Palaearctic, and Oriental Regions [21,22,23,24].

**Family Pseudococcidae** Westwood, 1840

***Maconellicoccus hirsutus*** (Green, 1908)Material examined: São Vicente, Mindelo, 10.ii.2018, 6 ♀♀ on *Hibiscus* sp. (Malvaceae); Sal, Santa Maria, 12.ii.2018, 8 ♀♀ on *Euphorbia* sp. (Euphorbiaceae); 14.ii.2018, 16 ♀♀ on *Tetraena fontanesi* (Zygophyllaceae)*; Heliotropium curassavicum* (Boraginaceae) and *Tamarix senegalensis* (Tamaricaceae).Distribution: A polyphagous species widespread throughout southern Asia, Australia, and Africa [20].***Palmicultor palmarum*** (Ehrhorn, 1916)Material examined: Sal, Santa Maria, 5.ii.2018, 17.ii.2018, 11 ♀♀ on palms; Santo Antão, 6.ii.2018, 5 ♀♀ on an unidentified plant. Distribution: Widely distributed in the Australasian, Oriental, and Palaearctic regions. It has also been reported from the Nearctic and Neotropics [1,11,18,25,26,27]. This is the first report from the Afrotropical Region.***Pseudococcus comstocki*** (?) (Kuwana, 1902)Material examined: Sal, Santa Maria, 14.ii.2018, 1**♀** on an unidentified plant.Distribution: A polyphagous species widely distributed in the Nearctic and Palaearctic regions and also reported from a few (1–6) countries in the Neotropical, Australasian, Oriental, and Afrotropical regions [1]. 

## 4. Discussion

Following the current study, 48 scale insect species are known from the Cape Verde Islands: 18 Diaspididae, 11 Pseudococcidae, 13 Coccidae, 2 Asterolecaniidae, 2 Monophlebidae, 1 Dactylopiidae, and 1 Ortheziidae (Table 1). This updated checklist includes all the scale insect species recorded from the Cape Verde Islands according to ScaleNet [1], plus the eight species reported here for the first time. However, *Orthezia urticae* (L.) is erroneously recorded in the ScaleNet catalogue, citing Van Harten et al. [9], but neither they nor any of the other authors who have studied the fauna of these islands, such as Fernandes [4,5,6,7] and Schmutterer et al. [8], mention it as being present. The list above also includes a further three diaspidid species (*Aonidomytilus albus* (Cockerell), *Duplachionaspis natalensis* (Maskell), and *Pinnaspis buxi* (Bouché)) which have been omitted from the ScaleNet database, although there are published records in Schmutterer et al. [8] and Van Harten et al. [9]. 

In addition to the new *Mirococcus* species described above, there may be other undescribed species in the various lots of material previously collected on these islands. Van Harten et al. [9] noticed an undescribed species of *Planococcoides* (which would now be placed in *Formicococcus*), which he considered had probably been introduced from mainland Africa. Van Harten et al. [9] also collected a *Ceroplastes* species, which they considered similar to but different from *C. rusci* (L.). It may have been these specimens from the Cape Verde Islands that were seen by Hodgson and Peronti [28], who also considered the specimens not to be *C. rusci* (L.) but possibly referable to one of the other cryptic species in this group.

Apart from *M. capeverdensis,* the newly recorded scale insects in the present study are all widely distributed, polyphagous species and considered to be plant pests. Thus, the cosmopolitan *Aulacaspis tubercularis* is a serious pest of mangos in many parts of the world [34,35,36]; *Hemiberlesia cyanophylli* causes damage to various ornamentals and avocado trees, while *Pseudaonidia trilobitiformis* was recorded as a pest of cocoa in The Democratic Republic of the Congo [1]. Another species, *Icerya aegyptiaca,* is considered a pest of *Artocarpus altilis* (Breadfruit) in the Pacific Region [23]. In addition, all the newly recorded mealybug species are known to cause some damage; thus, *Maconellicoccus hirsutus* is a potentially invasive pest of several crops, particularly pineapple [1]. *Palmicultor palmarum* attacks the leaves of oil palm and can kill young germinating plants [1,11], and *Pseudococcus comstocki* is a pest of many fruit and ornamental trees in the U.S.A. and Japan [1]. Although none of the above mentioned species were very abundant, each could potentially be important both ecologically and economically on the Cape Verde Islands.

## Figures and Tables

**Figure 1 insects-13-00999-f001:**
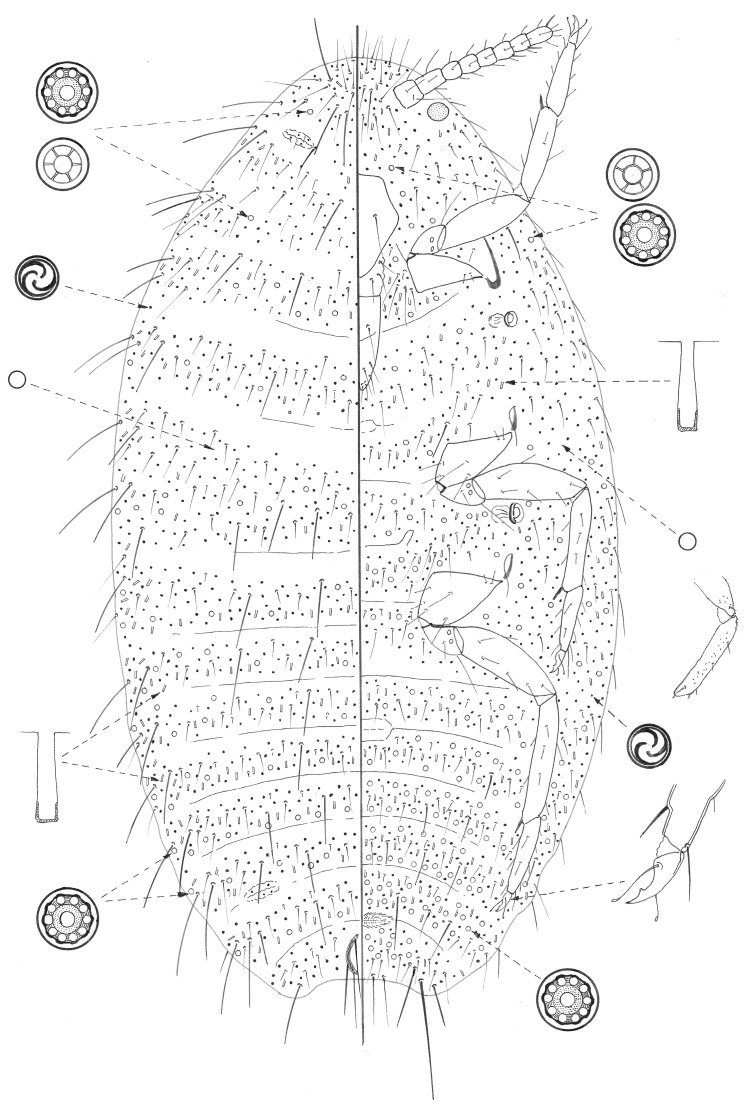
Adult female of *Mirococcus capeverdensis* Łagowska and Hodgson sp.n.

**Table 1 insects-13-00999-t001:** Checklist of scale insect species from the Cape Verde archipelago. All records from the present study are included in bold.

Family	Species	Cape Verde Islands	Host Plants	References
Asterolecaniidae	*Bambusaspis bambusae* (Boisduval)	Santiago	Bamboo	[7,9]
	*Russellaspis pustulans* (Cockerell)	Fogo; Santiago; **Sal**	*Ficus gnaphalocarpa*; *Trichilia emetica*; *Prosopis juliflora*; *Nerium oleander*; ***Tamarix senegalensis***	[6,9] **this publication**
Coccidae	*Ceroplastes rusci* (Linnaeus)	Fogo	*Ficus gnaphalocarpa?*	[5,9]
	*Ceroplastes stellifer* (Westwood)	**Santo Antão**	***Nerium oleander***; *Mangifera* sp.	[8,9] **this publication**
	*Ceroplastes* sp.	Santiago	*Codiaeum variegatum*	[9,28]
	*Coccus hesperidum* Linnaeus	Fogo; Santiago; São Nicolau; **Santo Antão; Sal**	*Ficus religiosa*; limon tree; bamboo; *Agave sisalana*; *Papaya* sp.; *Punica granatum*; *Terminalia catappa*; *Caylusea hexagyna*; *Plantago* sp., ***Cajanus cajan***; ***Schinus mole***; ***Heliotropium curassavicum***; ***Datura*** sp.	[4,6,7,8,9] **this publication**
	*Coccus longulus* (Douglas)	São Vicente; Santiago; Santo Antão	*Terminalia catappa*; *Codiaeum variagatum*; *Acacia* sp.	[9]
	*Coccus viridis* (Green)	Fogo; Santiago	Coffee; orange tree; lemon tree; *Anacardium occidentale*; *Mangifera indica*; *Gomphrena globata*; *Punica granatum*; date palm	[4,8,9,29]
	*Eucalymnatus tessellatus* (Signoret)	Santo Antão	*Mangifera indica*; *Cocos nucifera*	[8]
	*Parasaissetia nigra* (Nietner)	Fogo; Santiago; **Santo Antão**; São Nicolau	*Manihot* sp.; *Prunus dulcis*; *Euphorbia tuckeyana*; *Ficus thonningii*; ***Ficus*** sp.; *Parkinsonia aculeata*; *Terminalia catappa*; *Furcraea gigantea*; cassava; jojoba	[6,9,29] **this publication**
	*Pulvinaria aethiopica* (De Lotto)	Santiago	*Coffea* sp.	[4]
	*Pulvinaria elongata* (De Lotto)	Fogo; Santo Antão	Sugarcane	[9]
	*Pulvinaria psidii* Maskell	Santiago	*Ficus thonningii*	[9]
	*Pulvinaria urbicola* Cockerell	Brava; Santiago; **Santo Antão; Sal**	*Lantana camara*; sweet potato; *Achyranthes asper*; *Chenopodium* sp.; *Capsicum annuum*; *Artemisia gorgonum*; jacaranda; guava tree; ***Eranthemum purpurascens?***; ***Heliotropium curassavicum***; ***Nerium oleander***	[9] **this publication**
	*Saissetia coffeae* (Walker)	Santo Antão; Santiago; São Vicente	*Persea gratissima*; *Cycas* sp.; ornamental plants	[8,9]
	*Saissetia miranda* (Cockerell and Parrott)	Santiago	*Terminalia catappa*	[9]
Dactylopiidae	*Dactylopius opuntiae* (Cockerell)	Santiago	*Opuntia* sp.	[9]
Diaspididae	*Aonidomytilus albus* (Cockerell)	Brava; Santiago	Cassava	[8,9]
	*Aspidiotus destructor* Signoret	Brava	Coconut	[4]
	*Aspidiotus nerii* Bouché	Santiago	*Nerium oleander*; *Sideroxylon marmulana*	[9,29]
	*Aulacaspis rosae* (Bouché)	Cape Verde	*Rosa* sp.; *Rubus* sp.; other Rosaceous plants	[30,31,32]
	*Aulacaspis tubercularis* Newstead	**Santo Antão**	** *Mangifera indica* **	**this publication**
	*Chrysomphalus dictyospermi* (Morgan)	Santo Antão; Santiago	Avocado; lemon; banana; bamboo; coconut; *Punica granatum*; *Plantago* sp.	[8,9]
	*Diaspis echinocacti* (Bouché)	Santiago	*Opuntia ficus-indica*	[9]
	*Duplachionaspis natalensis* (Maskell)	Santiago	Coconut; *Arundo donax*	[8,9]
	*Fiorinia fioriniae* (Targioni Tozzetti)	Fogo; Santiago; Santo Antão	*Cocos nucifera*; avocado; mango; cashew tree; vine	[8,9]
	*Hemiberlesia cyanophylli* (Signoret)	**Sal**	***Tetraena fontanesii***; ***Heliotropium curassavicum***; ***Synedrella nodiflora***	**this publication**
	*Hemiberlesia lataniae* (Signoret)	**Sal**; Santiago; São Vicente; **Santo Antão**	*Brassica* sp.; guava; *Chenopodium* sp.; avocado; *Ipomoea tuberculata*; *Nerium oleander*; *Ficus carica*; *Terminalia catappa*; vine; jojoba; coconut; *Morus alba*; mango; date palm; *Cycas* sp.; ***Acacia saligna***	[4,5,8,9] **this publication**
	*Ischnaspis longirostris* (Signoret)	Santiago; Fogo; Santo Antão	Coconut; coffee; date palm; *Trichelia emetica*; mango	[7,8,9]
	*Lepidosaphes beckii* (Newman)	Santiago	*Citrus* sp.	[4,5,8,9,33]
	*Odonaspis saccharicaulis* (Zehntner)	Santiago	Sugar cane	[8]
	*Pinnaspis buxi* (Bouché)	Santiago	*Cyperus papyrus*	[9]
	*Pinnaspis strachani* (Cooley)	Brava; Santiago; **São Vicente**; Maio; Santo Antão; **Sal**	*Hevea brasiliensis*?, sweet potato; *Ziziphus mauritiana*; *Ipomoea tuberculata*; pigeonpea; *Cycas* sp.; ***Tetranea fontanesii***; ***Heliotropium curassavicum***	[4,6,8,9] **this publication**
	*Pseudaonidia trilobitiformis* (Green)	**São Vicente**	** *Nerium oleander* **	**this publication**
	*Radionaspis indica* (Marlatt)	Santiago	Mango	[7]
Monophlebidae	*Icerya aegyptiaca* (Douglas)	**Sal**	***Hibiscus*** sp.; ***Euphorbia*** sp.; ***Tamarix senegalensis***	**this publication**
	*Icerya purchasi* Maskell	Santiago; **Santo Antão**; Fogo; **São Vicente**; **Sal**	***Cajanus cajan***; *Parkinsonia aculeata*; *Ficus gnaphalocarpa*; *Artemisia gorgonum*; vine; ***Synedrella nodiflora***; ***Hibiscus*** sp.; ***Tamarix senegalensis***; ***Gossypium*** sp.	[6,8,9,29] **this publication**
Ortheziidae	*Insignorthezia insignis* (Browne)	Brava; Santiago	*Lantana camara*; *Amaranthus spinosus*	[6,9]
Pseudococcidae	*Mirococcus capeverdensis* Łagowska and Hodgson	**Sal**	** *Campylanthus glaber* **	**this publication**
	*Dysmicoccus boninsis* (Kuwana)	Santiago; **Santo Antão**	Sorghum; **sugar cane**	[8,9] **this publication**
	*Dysmicoccus brevipes* (Cockerell)	Sal; Santiago	Cyperaceae; pineapple	[8,9]
	*Ferrisia virgata* (Cockerell)	Brava; **Sal**; Santiago; **Santo Antão**	*Manihot esculenta*; *Phaseolus* sp.; *Psidium guajava*; sweet potato; *Citrullus colocynthus*; *Terminalia catappa*; *Ficus gnaphalocarpa*; *Anona muricata*; *Codiaeum variegatum*; ***Cajanus cajan***; ***Nerium oleander***	[8,9] **this publication**
	*Maconellicoccus hirsutus* (Green)	**Sal; São Vicente**	***Tetraena fontanesi***; ***Hibiscus* sp.**; ***Heliotropium curassavicum***; ***Tamarix senegalensis***; ***Euphorbia* sp.**	**this publication**
	*Palmicultor palmarum* (Ehrhorn)	**Sal; Santo Antão**	**Palm trees; unidentified plant**	**this publication**
	*Phenacoccus madeirensis* Green	Brava; Santiago; Santo Antão; São Nicolau	*Manihot esculenta*; *Lantana camara*; *Hibiscus rosa-sinensis*; *Acalypha hispida*; *Gossypium* sp.; *Artemisia gorgonum*; *Althaea rosea*; cotton; sugar-cane	[4,6,8,9]
	*Phenacoccus solani* Ferris	Santiago	*Amaranthus spinosus*	[9]
	*Planococcus citri* (Risso)	Brava; Fogo; Santiago; Santo Antão; **Sal**	*Coffea arabica*; *Citrus* sp.; *Cucurbita* sp.; sweet potato; *Zizyphus mauritiana*; *Solanum* sp.; pigeon pea; *Furcraea gigantea*; *Ficus thonningii*; *Mangifera* sp.; *Leucaena leucocephala*; *Caylusea hexagyna*; *Rumex* sp.; *Artocarpus incisa*; “Cruciferae”; ***Persea americana***; ***Datura*** sp.; ***Hibiscus*** sp.; ***Nerium oleander***	[7,8,9] **this publication**
	*Planococcoides* sp.	Santiago	*Ficus gnaphalocarpa*	[9]
	*Pseudococcus comstocki* (Kuwana)	**Sal**	**Unidentified plant**	**this publication**
	*Pseudococcus longispinus* (Targioni Tozzetti)	Santo Antão	*Artocarpus incisa*	[9]

## Data Availability

All the required data relevant to the presented study are included in the manuscript.

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
