# Peer review of "A New Species of the Mealybug Genus Mirococcus (Hemiptera: Coccomorpha: Pseudococcidae) from the Cape Verde Islands, with New Records and an Updated Checklist of Scale Insect Species"

_insects, 2022, doi:10.3390/insects13110999_

Round 1

Reviewer 1 Report

Comments to: A new species of the mealybug genus Mirococcus (Hemiptera:

Coccomorpha: Pseudococcidae) from the Cape Verde Islands,

It is a very good paper written by well-known specialists. The new species is correctly described and well-illustrated and a key to similar African species is included. Moreover, an updated checklist of the species presently known in the Cape Verde islands, including some new records, is reported.

I noticed few typing errors mostly on the Latin names of the host plants and added very few suggestions.

Line 181: the correct name of the species is hockingi.

Line 217: Humococcus (Mirococcopsis) mackenziei: a new comb.? In Scalenet: Humococcus mackenziei, in Danzig & Gavrilov: Mirococcopsis mackenziei.

Line 266: Morrison, 1927 not in bold

Line 288: Pseudococcus comstocki (?): doubtful identification?

Lines 304-305: “Including the two so far unidentified species [9], this brings the total number of species known on the Cape Verde Islands to 50”: may I suggest deleting this sentence. You clearly explain the same concept in the following 306 – 312 lines.

Line 311: “The list below also includes…: in the present layout the list is above.

Line 321: in my opinion you can delete “serious”, referred to P. comstocki.

TABLE:  

Corrections:

Coccus hesperidum, host plant: please correct Agava to Agave

Eucalymnatus tessellatus:  not in bold. Please add a parenthesis after Signoret

Pulvinaria elongate: the correct name is elongata

Saissetia miranda: please delete ;

Icerya aegyptiaca: please, correct Tamari to Tamarix

Planococcus citri: please, correct Leucena to Leucaena

Hemiberlesia lataniae: Brassica sp. in italics

There is no caption for the table. I think it can be useful clarifying in the caption that the new collecting places and the newly recorded host plants are reported in bold.

Comments to: A new species of the mealybug genus Mirococcus (Hemiptera:

Coccomorpha: Pseudococcidae) from the Cape Verde Islands,

It is a very good paper written by well-known specialists. The new species is correctly described and well-illustrated and a key to similar African species is included. Moreover, an updated checklist of the species presently known in the Cape Verde islands, including some new records, is reported.

I noticed few typing errors mostly on the Latin names of the host plants and added very few suggestions.

Line 181: the correct name of the species is hockingi.

Line 217: Humococcus (Mirococcopsis) mackenziei: a new comb.? In Scalenet: Humococcus mackenziei, in Danzig & Gavrilov: Mirococcopsis mackenziei.

Line 266: Morrison, 1927 not in bold

Line 288: Pseudococcus comstocki (?): doubtful identification?

Lines 304-305: “Including the two so far unidentified species [9], this brings the total number of species known on the Cape Verde Islands to 50”: may I suggest deleting this sentence. You clearly explain the same concept in the following 306 – 312 lines.

Line 311: “The list below also includes…: in the present layout the list is above.

Line 321: in my opinion you can delete “serious”, referred to P. comstocki.

TABLE:  

Corrections:

Coccus hesperidum, host plant: please correct Agava to Agave

Eucalymnatus tessellatus:  not in bold. Please add a parenthesis after Signoret

Pulvinaria elongate: the correct name is elongata

Saissetia miranda: please delete ;

Icerya aegyptiaca: please, correct Tamari to Tamarix

Planococcus citri: please, correct Leucena to Leucaena

Hemiberlesia lataniae: Brassica sp. in italics

There is no caption for the table. I think it can be useful clarifying in the caption that the new collecting places and the newly recorded host plants are reported in bold.

Comments to: A new species of the mealybug genus Mirococcus (Hemiptera:

Coccomorpha: Pseudococcidae) from the Cape Verde Islands,

It is a very good paper written by well-known specialists. The new species is correctly described and well-illustrated and a key to similar African species is included. Moreover, an updated checklist of the species presently known in the Cape Verde islands, including some new records, is reported.

I noticed few typing errors mostly on the Latin names of the host plants and added very few suggestions.

Line 181: the correct name of the species is hockingi.

Line 217: Humococcus (Mirococcopsis) mackenziei: a new comb.? In Scalenet: Humococcus mackenziei, in Danzig & Gavrilov: Mirococcopsis mackenziei.

Line 266: Morrison, 1927 not in bold

Line 288: Pseudococcus comstocki (?): doubtful identification?

Lines 304-305: “Including the two so far unidentified species [9], this brings the total number of species known on the Cape Verde Islands to 50”: may I suggest deleting this sentence. You clearly explain the same concept in the following 306 – 312 lines.

Line 311: “The list below also includes…: in the present layout the list is above.

Line 321: in my opinion you can delete “serious”, referred to P. comstocki.

TABLE:  

Corrections:

Coccus hesperidum, host plant: please correct Agava to Agave

Eucalymnatus tessellatus:  not in bold. Please add a parenthesis after Signoret

Pulvinaria elongate: the correct name is elongata

Saissetia miranda: please delete ;

Icerya aegyptiaca: please, correct Tamari to Tamarix

Planococcus citri: please, correct Leucena to Leucaena

Hemiberlesia lataniae: Brassica sp. in italics

There is no caption for the table. I think it can be useful clarifying in the caption that the new collecting places and the newly recorded host plants are reported in bold.

Author Response

Subject: Response to comments raised by reviewer

Manuscript ID: insects-1979430 title: "A new species of the mealybug genus Mirococcus (Hemiptera: Coccomorpha: Pseudococcidae) from the Cape Verde Islands, with new records and an updated checklist of scale insect species"

Dear Sir/Madam,

While thanking you for very valuable comments and suggestions made on the MS, I would like to submit the revised Ms along with responses to reviewers’ comments which are listed below for your kind attention. 

Reviewer 1:

General comments:
Line 181: the correct name of the species is hocking: We are agree with reviewer, in this part of manuscript we made a mistake, this has been corrected

Line 217: Humococcus (Mirococcopsis) mackenziei: a new comb.? In Scalenet: Humococcus mackenziei, in Danzig & Gavrilov: Mirococcopsis mackenziei: these two names of one species are use and we decided don’t change these in the text.

Line 266: Morrison, 1927 not in bold: this has been corrected

Line 288: Pseudococcus comstocki (?): doubtful identification? Yes, the question mark in brackets indicates that the species identification is doubtful

Lines 304-305: “Including the two so far unidentified species [9], this brings the total number of species known on the Cape Verde Islands to 50”: may I suggest deleting this sentence. You clearly explain the same concept in the following 306 – 312 lines.: Yes, we are agree with the reviewer, it was deleted

Line 311: “The list below also includes…: in the present layout the list is above: this has been corrected below into above

Line 321: in my opinion you can delete “serious”, referred to P. comstocki: this word was deleted in the text

TABLE:  We are agree with reviewer, in this part of manuscript we made a lot of mistakes, all of these listed below were corrected in the table

Corrections:

Coccus hesperidum, host plant: please correct Agava to Agave

Eucalymnatus tessellatus:  not in bold. Please add a parenthesis after Signoret

Pulvinaria elongate: the correct name is elongata

Saissetia miranda: please delete ;

Icerya aegyptiaca: please, correct Tamari to Tamarix

Planococcus citri: please, correct Leucena to Leucaena

Hemiberlesia lataniae: Brassica sp. in italics

There is no caption for the table. I think it can be useful clarifying in the caption that the new collecting places and the newly recorded host plants are reported in bold: Yes, we are agree with the reviewer, it was corrected, this information was introduced.

Table: This part of ms has been carefully re-editing and corrected. To make the table more readable and to avoid split of the latin name of scale insects and host plants, according to our opinion this Table should be layed out on a Landscape page and then spread across the page and then it would not be necessary to split any of the names – we corrected it.

All reviewer comments have been included in this revised submission.

Yours sincerely

Katarzyna Golan

Reviewer 2 Report

This research is valuable to be published because it contains updated information including new species/records of scale insects. However, the authors should make clear several points I raised especially in the description of new species and the taxonomic key in the manuscript. Further editing/formatting for the other parts of the manuscript is also needed for publication. Please find specific comments in the attached.

Author Response

Subject: Response to comments raised by Reviewer

Manuscript ID: insects-1979430 title: "A new species of the mealybug genus Mirococcus (Hemiptera: Coccomorpha: Pseudococcidae) from the Cape Verde Islands, with new records and an updated checklist of scale insect species"

Dear Sir/Madam,

While thanking you for very valuable comments and suggestions made on the MS, I would like to submit the revised Ms along with responses to reviewers’ comments which are listed below for your kind attention.  All comments have been included in this revised submission.

Reviewer 2:

This research is valuable to be published because it contains updated information including new species/records of scale insects. However, the authors should make clear several points I raised especially in the description of new species and the taxonomic key in the manuscript. Further editing/formatting for the other parts of the manuscript is also needed for publication. Please find specific comments in the attached.

Line 13: we changed, corrected and adjusted in the text of manuscript the scientific names of scale insects and plants

Line 14: According to reviewer comment we decided to change the sentence “A taxonomic key to the mealybugs from the Afrotropical Region is provided” into ” A key to the mealybugs from the Afrotropical Region that lack cerarii is provided” We hope this sentence will be more precise for readers

Line 15: the sentence with: among them an undescribed species is correct, we did’t change it

Line 69: we changed figures unto figure, of course it should be singular

Line 92 and Reviewer comments about key to mealybug: In our opinion the key in this manuscript works well.  Introducing additions to the key as requested by the reviewer would take a long time, but according to the authors, it is not necessary because the key works well in its current form.

Line 104: ‘deposited BMNH’ was change to deposited in BMNH

Line 115: according to authors the sentence ‘Ostioles present, with few trilocu116 lar pores on lips and no setae’ is correct.

Line 117: we added more details according to reviewer suggestion: „each up to 60 µm long” changed to „lengths ranging from 35 µm to 60 µm”

Line 120: we add at end of line after „µm long “ information about distribution of the seta on dorsum acording to the comments: „with a few on all segments.”

Line 124: we agree with reviewer and insert after „smaller pores”.: „Quinquelocular pores otherwise absent”.

Line 144: marginally was correct into margin

Line 145: this sentence has been corrected into : Of the species currently included in Mirococcus, adult female M. capeverdensis sp. n. is most similar to that of M. inermis (Hall)….”

Line 160: We decided not to change this sentence. According to our opinion this is correct

Line 172: Insert after bouhelieri (Goux)  and put Morocco,in brackets Giving: bouhelieri (Goux) (Morocco),  Insert after dactyloni (Bodenheimer) and put Tunisia in brackets Giving: dactyloni (Bodenheimer) (Tunisia)

 Line 218: we changed of to or in the sentence: „Translucent pores either absent of more wide spread on hind legs  into „Translucent pores either absent or more widespread on hind legs”

Line 292: according to comments we changed the heading of the Right hand column -  Authors to References

Line 298:  „….plus the eight new species records reported here for the first time” into plus the eight species reported here for the first time”

Table: This part of ms has been carefully re-editing and corrected. To make the table more readable and to avoid split of the latin name of scale insects and host plants, according to our opinion this Table should be layed out on a Landscape page and then spread across the page and then it would not be necessary to split any of the names – we corrected it.

Yours sincerely

Katarzyna Golan